# Adaptive Integral Sliding Mode Based Course Keeping Control of Unmanned Surface Vehicle

José Antonio González-Prieto [1,*], Carlos Pérez-Collazo [1,*] and Yogang Singh [2]

1   Defense University Center, Spanish Naval Academy, Plaza de España, s/n, 36920 Marin, Galicia, Spain
2   Industrial Systems Engineering (ISyE), FlandersMake@Ghent University, Graaf Karel de Goedelaan 2B, Geb.A, 8500 Kortrijk, Belgium; yogang.singh@ugent.be
*   Correspondence: jose.gonzalez@cud.uvigo.es (J.A.G.-P.); carlos.perez.collazo@cud.uvigo.es (C.P.-C.)

**Abstract:** This paper investigates the course keeping control problem for an unmanned surface vehicle (USV) in the presence of unknown disturbances and system uncertainties. The simulation study combines two different types of sliding mode surface based control approaches due to its precise tracking and robustness against disturbances and uncertainty. Firstly, an adaptive linear sliding mode surface algorithm is applied, to keep the yaw error within the desired boundaries and then an adaptive integral non-linear sliding mode surface is explored to keep an account of the sliding mode condition. Additionally, a method to reconfigure the input parameters in order to keep settling time, yaw rate restriction and desired precision within boundary conditions is presented. The main strengths of proposed approach is simplicity, robustness with respect to external disturbances and high adaptability to static and dynamics reference courses without the need of parameter reconfiguration.

**Keywords:** unmanned surface vehicle; Guidance, Navigation and Control; course keeping; adaptive sliding mode

## 1. Introduction

With the growing advancement in the sensor technology and navigation aids, USVs are becoming a popular tool in maritime domain for several applications ranging from environmental monitoring, military surveillance to scientific surveying, and data collection. Mission oriented approach of USVs subject them to several types of maritime environment comprising of wind, wave, and sea surface currents leading to requirement of designing and developing several autonomy levels for successful operation. Henceforth, design and development of approaches for Guidance, Navigation, and Control (GNC) of a USV is an important research area for constructing operational and tactical approaches for seven different operational autonomy level of USVs as described by International Maritime Organisation (IMO).

Guidance and control of USV plays an important role in motion control system to manipulate the forces to enable a USV to follow a desired path whilst maintaining the stability. Three approaches, namely, waypoint control, path following control and trajectory tracking are generally considered in the domain of marine robotics to enable a USV to follow a designated path [1,2]:

- **Waypoint control:** In this strategy, Line of Sight (LOS) based approach is adopted to follow a certain waypoints, generated heuristically, in the required maritime environment.
- **Path following control:** In this strategy, a path generated through path planning algorithms is used as a reference, to be followed with no temporal constraints. Here, USV should converge and follow the desired path without any time constraints and simultaneously satisfies its assigned velocity profile.
- **Trajectory tracking:** In this strategy, temporal constraints are enforced upon the path generated using path planners. This is predominantly used with fully actuated marine vehicles reasoned with better maneuvering capabilities.

Sailing conditions and unpredictability of environmental disturbances can have a significant impact on the ship's dynamics. It is therefore necessary to develop a nonlinear controller that overcomes unknown disturbances and ensures robustness. As long as parameter uncertainties and unknown bounded disturbances remain, the adaptive method is likely to remain a superior approach. It is intended that vessel steering autopilots will force the ship to follow a predetermined course with a fixed speed by controlling the rudder angle, creating a course keeping problem that the current study is attempting to resolve.

### 1.1. State of the Art

The problem of **course keeping control** is highly non-linear in nature and has been studied from a perspective of observed disturbance control using sliding mode control (SMC) approach. The SMC problem for USVs, subjected to, higher order non linear operational disturbances, have been studied with varying control approaches like sliding mode [3–6]; fuzzy sliding mode [7]; proportional derivative fuzzy [8]; backstepping [9–12]; backstepping with adaptive radial basis function neural network [13]; sine function-based non-linear feedback [14]; hyperbolic tangent based nonlinear control [15]; sigmoid based nonlinear control [16]; function adaptive neural path following control [17]; model predictive control [18,19]; event-triggered control approach [20] and non-linear feedback power functions [21].

In order to make control robust to disturbances and uncertainties, several approaches has been proposed in the SMC literature, see [22–32]. Some proposals of advanced sliding manifolds include recursive nonlinear sliding manifolds [33–35], adaptive integral sliding mode approach [36–38], non linear full order dynamics [39,40], sliding surfaces with adaptive damping parameters [41–43] and, in the last years, a vast collection of homogeneity based works, see [44] for instance. Applications of the properties of homogeneous systems is an important field of study in the current development of analysis and design of nonlinear controllers and observers. Homogeneity simplifies analysis and design of nonlinear control systems since the homogeneous vector fields have many properties similar to linear one and provides solutions with finite-time and fixed-time stability.

The dynamics generated by an homogeneous controller can be seen as a lineal dynamic system with an adaptive gain that grows to $\infty$ as $|x(t)| \rightarrow 0$, generating the well know singularity at the origin which is undesired for real applications. Nevertheless, as commented in [45], the practical implementation of homogeneous dynamics system designed in the continuous time domain prevents the use of explicit *Euler* discretization scheme to achieve a mere copy of the continuous time approach due to its simplicity. This type of discretization is considered inappropriate, especially when set-valued functions has to be considering, causing numerical chattering and sensitivity to the gains. As a result, without addressing the discretization issue, any comparison between homogeneous based solutions and other types of proposals may potentially lead to unfair conclusions.

Based on the aforementioned results, in order to keep the discretization process simple, an adaptive lineal sliding mode surface law, that includes a nested integral sliding surface is introduced in this work. In this case, the dynamics flows with adaptive and finite damper gain, avoiding the effects of the peaking transient response inherent to linear systems and allowing fast responses at steady state, approximating the behaviour obtained with homogeneous solutions.

### 1.2. Major Contributions

The paper makes following contributions to the current state of existing approaches to SMC techniques for USVs:

- A number of simulation studies in the manuscript demonstrate that the proposed adaptive control approach can be reconfigured for various input trajectories and marine environmental disturbances, without requiring parametric adjustment.
- The cut-off frequency of the system response is an indication of the bound to be assigned to the disturbance derivative in the algorithm. This relationship is based on low-pass filtering properties associated with the second order adaptive linear dynamics generated

at the sliding variable. As a result, frequencies over $\omega_c$ do not affect the sliding variable response. In practice, this feature offers some advantages when estimating the maximum value of the disturbance derivative is a challenging task.

- The proposed adaptive profile generates low/high gains based on the absolute error. As a result, the control input is not saturated when there is a large error (gain is small) and the response at steady state becomes fast disturbance compensation (gain is large).
- Based on the adaptive placement of two poles relating to a second order dynamical system with critical damping, we can generate an overdamped response that avoids the occurrence of considerable overshoots.
- By avoiding the need of the derivative of the fractional power terms with respect to time, the singularity problem associated with terminal sliding mode solutions can be avoided. Thus, the high sensitive performance around the equilibrium point generated by set value or fractional order functions can be reduced.

This paper has been structured as follows. First, in Section 2 we present the nonlinear dynamic model of the course keeping problem, the desired objectives to be achieved and a theoretical stability tool that is used in the posterior analysis of the control algorithm. Then, Section 3 describes the proposed adaptive integral sliding mode (AISM) algorithm. Results from numerical simulations are then presented and discussed in Section 4. Finally, conclusions are drawn in Section 5.

## 2. Problem Statement

The motion of the USV is shown in Figure 1, where a six degrees of freedom (DOF) model is presented. The earth fixed **Oo** is an inertial reference frame fixed to the earth's surface and the body fixed with origin **O** is a moving coordinate frame that it is fixed to the craft as in given in [1]. It is assumed an homogeneous mass distributed and xz-plane symmetrical, such that origin of the body fixed reference frame is chosen to be coincident with the center of gravity.

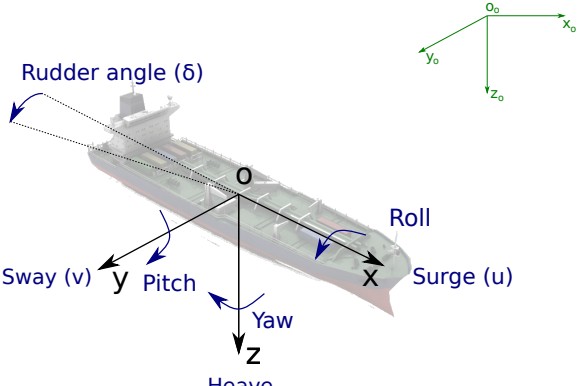

**Figure 1.** 6 DOF motion representation with North-East-Down coordinate system (green) and body fixed reference frame (black).

If we consider the course keeping problem the dynamics of heave, roll, and pitch can be neglected, so that the reduced model dynamics are given as

$$m(\dot{u} - vr - x_c r^2) = X \tag{1}$$
$$m(\dot{v} + ur + x_c \dot{r}) = Y \tag{2}$$
$$I_z \dot{r} + m x_c (\dot{v} + ur) = N \tag{3}$$

where $m$ is the mass, $u$ is the surge velocity, $v$ is the sway velocity, $r$ is the yaw rate, $I_z$ is the rotational inertia with respect to $z$ axis, $x_c$ is the $x$ coordinate of the vehicle center in the fixed body reference frame and $X$, $Y$ and $N$ are the external forces and moments with respect to the surge, sway, and yaw, respectively.

Assumption of constant forward speed and using the ship's *Norrbin* nonlinear mathematical model, see [46], implies that the steering equations of motion can be obtained as

$$
\dot{\psi}(t) = r(t)
$$
$$
\dot{r}(t) = f(r) + g\delta(t) + d(t) \tag{4}
$$

where, $\psi(t)$ is the yaw (orientation) angle, $r(t)$ is the yaw rate, $\delta(t)$ is the rudder angle (the control variable to be designed) and $d(t)$ is an unknown term to be compensated that includes parametric uncertainty and external disturbances (wind, waves, mobile loads). The dynamics functions are given as

$$
g = -\frac{K}{T}
$$
$$
f(r) = -\frac{K}{T}H(r)
$$
$$
H(r) = a_1 r + a_2 r^3 \tag{5}
$$

where $(K, T)$ are hydrodynamic coefficients and $(a_1, a_2)$ are *Norrbin* coefficients.

In the course keeping problem it is required that the yaw angle $\psi$ follows a reference angle $\psi_r$ by means of the design of the rudder control signal $\delta(t)$. The following assumptions are taking account in this work.

**Assumption 1.** *$d(t)$ in (4) satisfies the following restriction*

$$
|d(t)| \leq D
$$

*with $D > 0$ a positive unknown real number.*

**Assumption 2.** *$d(t)$ in (4) satisfies the following restriction*

$$
|\dot{d}(t)| \leq \Delta
$$

*with $\Delta > 0$ a positive known real number.*

**Assumption 3.** *Henceforth, it is assumed that a reference yaw establish the desired input to be tracked, which can be obtained by means of path planning algorithms, that account for different environment constraints as in [47–50]. A dynamic reference model is used, in this work, to generate the desired course $(\psi_r(t), \dot{\psi}_r(t), \ddot{\psi}_r(t))$.*

The objective is to design a control law that creates overdamped responses with minimal overshooting (undershooting) and robustness properties for response of the yaw error, which is defined as

$$
e(t) = \psi(t) - \psi_r(t) \tag{6}
$$

In order to check the control performance of the proposed controller for the course keeping problem, we consider the following performance analysis indices mentioned in [10,12].

$$
MAE = \frac{1}{t_\infty - t_0} \int_{t_0}^{\infty} |e(t)| dt \tag{7}
$$

$$
MIA = \frac{1}{t_\infty - t_0} \int_{t_0}^{\infty} |\delta(t)| dt \tag{8}
$$

$$
MTV = \frac{1}{t_\infty - t_0} \int_{t_0}^{\infty} |\delta(t) - \delta(t - \tau)| dt \tag{9}
$$

where $\tau$ is the sampling time used in the simulation.

Furthermore, to check out the robustness properties of the solution, we compare the results with the algorithms proposed in [10,12] applying the following conditions:

- As in [10,12], we test two problems that uses two different types of reference input signals: step and sinusoidal.
- The tests includes results without disturbances ($d(t) = 0$) and with disturbances ($d(t) \neq 0$).
- The algorithm parameters are configured in the case of the step input reference without disturbances, such that all solutions provide the same value of the MIA index at the end of the test time.
- After that, the algorithms parameters are fixed and tested in the case of step with disturbances and in the case of the sinusoidal input reference. In this way we check the robustness of the solutions with respect to its capacity of adaptation to different scenarios from a specific parameter configuration.

The following theorem is introduced in order to analyse the stability properties of the AISM proposed solution.

**Theorem 1.** *Consider the following cascade system*

$$\dot{z}_1 = f_1(t, z_1) + g_1(t, z_1, z_2)z_2 \tag{10}$$

$$\dot{z}_2 = f_2(t, z_2) \tag{11}$$

*where $z_1 \in \mathbb{R}^n$, $z_2 \in \mathbb{R}^m$, $f_1(t, z_1)$ is continuously differentiable in $(t, z_1)$, and $f_2(t, z_1)$ and $g_1(t, z_1, z_2)$ are continuous and locally Lipschitz in $z_2$ and $(z_1, z_2)$, respectively.*
*The dynamics of (10) when $z_2 = 0$ are*

$$\dot{z}_1 = f_1(t, z_1) \tag{12}$$

*If systems (12) and (11) are globally uniformly asymptotically stable (GUAS) and we know a $C^1$ Lyapunov function $V(t, z_1)$, two class-$K_\infty$ functions $\phi_1$ and $\phi_2$, a class-K $\phi_3$ function and a positive semidefinite function $W(z_1)$ such that*

$$\phi_1(||z_1||) \leq V(t, z_1) \leq \phi_2(||z_1||) \tag{13}$$

$$\frac{\partial V}{\partial t} + \frac{\partial V}{\partial z_1} f_1(t, z_1) \leq -W(z_1) \tag{14}$$

$$||\frac{\partial V}{\partial z_1}|| \leq \phi_3 \tag{15}$$

*Besides, for each fixed $z_2$ there exists a continuous function $\zeta : \mathbb{R}^+ \to \mathbb{R}^+$ such that*

$$\lim_{s \to \infty} \zeta(s) = 0 \tag{16}$$

$$||\frac{\partial V}{\partial z_1} g_1(t, z_1, z_2)|| \leq \zeta(||z_1||)W(z_1) \tag{17}$$

*Then we can conclude that the cascade system (10) and (11) is GUAS.*

**Proof.** See [51]. □

## 3. Adaptive Integral Sliding Mode Surface Control Design

Derivation of $e(t)$ in (6) leads to

$$\dot{e}(t) = r(t) - \dot{\psi}_r(t) \tag{18}$$

An adaptive sliding surface $s(t)$ variable is defined as

$$s(t) = \dot{e}(t) + \lambda(e)e(t) \tag{19}$$

with $\lambda(e)$ a real positive time varying parameter.

Consider the integral term $\bar{s}(t)$

$$\bar{s}(t) = \int_0^t s(t)dt \tag{20}$$

Let us choose the control law as

$$\delta(t) = \frac{1}{g}(-f(r) + \ddot{\psi}_r(t) - \lambda(e)\dot{e}(t) - \dot{\lambda}(e)e(t) - \alpha(s,\bar{s})s(t) - \gamma(e)\bar{s}(t)) \tag{21}$$

with $\lambda(e)$ defined as

$$\lambda(e) = \max(\lambda_{min}, \lambda_{max} - (\frac{\lambda_{max} - \lambda_{min}}{|e(0)|})|e(t)|), \tag{22}$$

the variable $z(t)$, related a new sliding surface, defined as

$$z(t) = s(t) + \frac{\alpha}{2}\bar{s}(t) \tag{23}$$

and the parameters $\alpha(s,\bar{s})$, $\gamma(\alpha)$ given as

$$\dot{\alpha}(s,\bar{s}) = \begin{cases} \kappa|z|^{\zeta}\operatorname{sign}(z)\operatorname{sign}(s) & \text{if } |z| > \mu \wedge \alpha_{min} < \alpha < \alpha_{max} \\ 0 & \text{otherwise} \end{cases} \tag{24}$$

$$\gamma(\alpha) = \frac{\alpha^2}{4} \tag{25}$$

with $\zeta(e)$ defined as

$$\zeta(e) = (\frac{\zeta_{max} - \zeta_{min}}{|e(0)|})|e(t)| + \zeta_{min} \tag{26}$$

and $\mu$

$$\mu = \sqrt[\zeta+1]{\frac{\Delta}{\kappa}} \tag{27}$$

Derivation of $\gamma(\alpha)$ and $\lambda(e)$ are given as

$$\dot{\gamma}(\alpha) = \frac{\alpha}{2}\dot{\alpha} \tag{28}$$

$$\dot{\lambda}(e) = \begin{cases} -(\frac{\lambda_{max}-\lambda_{min}}{|e(0)|})\operatorname{sign}(e(t))\dot{e}(t) & \text{if } \lambda > \lambda_{min} \\ 0 & \text{if } \lambda \leq \lambda_{min} \end{cases} \tag{29}$$

The control algorithm is designed by an appropriate selection of the parameters $\alpha_{max}$, $\alpha_{min}$, $\lambda_{max}$, $\lambda_{min}$, $\alpha(0)$, $\kappa$, $\zeta_{max}$ and $\zeta_{min}$, as it will be introduced in the numerical simulations section. Figure 2 show the control loop and the detail of the block diagram structure of the course keeping algorithm.

**Theorem 2.** *Consider the ship course dynamics described in (4) that complies with assumption 2. The application of the control law (21) to dynamic system (4) implies that the closed compact set $\Omega_e$ defined as*

$$\Omega_e = \{(e(t), \dot{e}(t)) \in \mathbb{R}^2 : |e(t)| < \frac{\mu}{|\cos(\theta)||\sin(\vartheta)|} \wedge |\dot{e}(t)| < \frac{\mu}{|\cos(\theta)||\cos(\vartheta)|}\} \tag{30}$$

*is GUAS with $\mu$, $\theta$ and $\vartheta$ given as*

$$\theta = \text{atan}(\lambda) \tag{31}$$

$$\vartheta = \text{atan}(\frac{\lambda}{2}) \tag{32}$$

**Proof.** Application of control law (21) to dynamic system (4) creates the following cascade system.

$$\dot{e}(t) = -\lambda e(t) + s(t) \tag{33}$$

$$\dot{s}(t) = -\alpha s(t) - \gamma \bar{s}(t) + d(t) \tag{34}$$

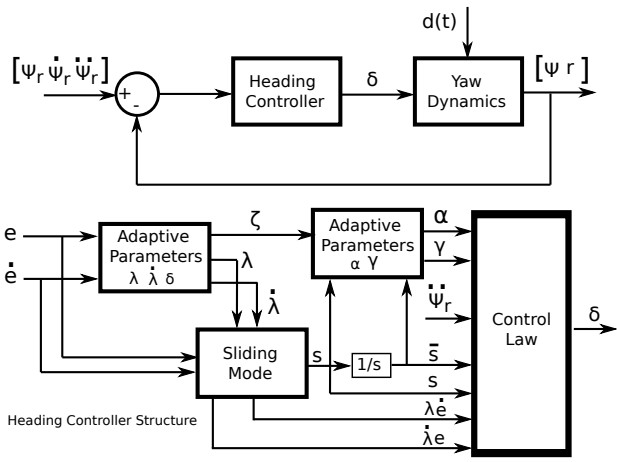

**Figure 2.** Course keeping control system and detail of heading control law block diagram.

The dynamics of $\dot{e}(t)$ when $s(t) = 0$ (dynamics of the yaw error at the sliding condition) are

$$\dot{e}(t) = -\lambda e(t) \tag{35}$$

with $\lambda > 0$. Therefore system (35) is GUAS, with exponential convergence.

Derivation of $\dot{s}(t)$ leads to

$$\ddot{s}(t) + \alpha \dot{s}(t) + \gamma s(t) + \dot{\alpha} s(t) + \dot{\gamma} \bar{s}(t) + \dot{d}(t) = 0$$

From (25), (28) and (23) it is obtained

$$\ddot{s}(t) + \alpha \dot{s}(t) + \frac{\alpha^2}{4} s(t) + \dot{\alpha} z(t) + \dot{d}(t) = 0$$

- **Case 1:** $|z| \leq \mu \wedge \alpha = \alpha_{min} \wedge \alpha = \alpha_{max}$
  Assuming the worst case scenario, that is, when $z > \mu \wedge \alpha = \alpha_{min}$, substitution of $\dot{\alpha} = 0$ implies that the second order dynamics related to $s(t)$ are

$$\ddot{s}(t) + \alpha_{min} \dot{s}(t) + \frac{\alpha_{min}^2}{4} s(t) + \dot{d}(t) = 0 \tag{36}$$

Let's define a sliding vector state $\eta(t)$ as

$$\eta(t) = \begin{bmatrix} s(t) & \dot{s}(t) \end{bmatrix}^T \tag{37}$$

Dynamics of $\eta(t)$ are given by

$$\dot{\eta}(t) = A\eta(t) + F(t) \tag{38}$$

with

$$A = \begin{pmatrix} 0 & 1 \\ -\alpha_{min} & -\frac{\alpha_{min}^2}{4} \end{pmatrix} \tag{39}$$

and

$$F(t) = \begin{pmatrix} 0 \\ \dot{d}(t) \end{pmatrix} \tag{40}$$

Lets' define

$$p_1 = \frac{\alpha_{min}}{8} + \frac{2(\alpha_{min} + 1)}{\alpha_{min}^2} \tag{41}$$

$$p_2 = \frac{0.5}{\alpha_{min}} \tag{42}$$

$$p_3 = \frac{2(\alpha_{min} + 1)}{\alpha_{min}^3} \tag{43}$$

and P a symmetric positive definite matrix

$$P = \begin{pmatrix} p_1 & p_2 \\ p_2 & p_3 \end{pmatrix} \tag{44}$$

with determinant

$$|P| = \frac{\lambda^4 + 16\lambda^2 + 32\lambda + 16}{4\lambda^5} > 0 \tag{45}$$

It can be shown that

$$PA + A^T P = -Q \tag{46}$$

where Q is the identity matrix of size 2 × 2. Therefore, the selection of a Lyapunov candidate function

$$V(\eta) = \frac{1}{2}\eta^T P\eta \tag{47}$$

leads to

$$\dot{V}(\eta) = -\eta^T Q\eta + F^T P\eta \tag{48}$$

$$= -\eta^T Q\eta - \dot{d}(t)(p_2 s(t) + p_3 \dot{s}(t)) \tag{49}$$

Let's define

$$p^* = \sqrt{2}\max(p_2, p_3) \tag{50}$$

Therefore

$$|\dot{d}(t)(p_2 s(t) + p_3 \dot{s}(t))| \leq p^* \Delta ||\eta|| \tag{51}$$

Applying (51) and assumption 2 it is obtained

$$\dot{V}(\eta) < -\gamma_Q^{min}||\eta||^2 + p^* \Delta ||\eta|| \tag{52}$$

Note that $\gamma_Q^{min} = 1$, thus the closed set $\Omega_\eta$, which includes the origin, defined as

$$\Omega_\eta = \{\eta(t) \in \mathbb{R}^2 : ||\eta|| \le p^*\Delta\} \tag{53}$$

is GUAS with exponential convergence (see [52]). The values of $p^*$ and $\Delta$ determines the size of the stable closed set, so that this condition limits how the algorithm may be applied. Inside $\Omega_\eta$ there are two possible cases

- $\text{sign}(s\dot{s}) < 0$: implies that $|s(t)| \to 0$, that is, the dynamics is stable an converges to the origin, with a value of $\alpha$ adjusted to keep this condition.
- $\text{sign}(s\dot{s}) \ge 0$: implies that $|s(t)|$ grows inside $\Omega_\eta$, that is, with an upper bound, or its value is stationary. According to (23) and (1), there exist an instant where the condition $\text{sign}(z)\,\text{sign}(s) > 0$ is met, which implies that $\alpha$ grows, that is, condition $\dot{\alpha} \ne 0$ is achieved. Therefore, $\alpha$ grows only when it is needed to keep the sliding mode condition at steady state, which is related to the performance given by the value of $\lambda_{max}$.

- **Case 2:** $|z| > \mu \wedge \alpha_{min} < \alpha < \alpha_{max}$
Substitution of $\dot{\alpha} \ne 0$ from (24) implies that the second order dynamics equation related to $s(t)$ is

$$\ddot{s}(t) + \alpha\dot{s}(t) + \frac{\alpha^2}{4}s(t) + \kappa|z|^{\zeta+1}\,\text{sign}(s) + \dot{d}(t) = 0 \tag{54}$$

Applying assumption 2 and condition $|z| > \mu$ implies that

$$\kappa|z|^{\zeta+1}\,\text{sign}(s) + \dot{d} = \rho_z(t)s \tag{55}$$

with $\rho_z(t) > 0$. Accordingly, because of assumption 2, the characteristic polynomial of (54) is *Hurwitz* for all $z(t) \notin \Omega_z$ where

$$\Omega_z = \{z(t) \in \mathbb{R} : |z(t)| < \mu\} \tag{56}$$

with $\mu$ defined in (27). This implies that (34) is GUAS with respect to the closed set $\Omega_z$. Note that dynamics in (54) can be viewed as a second order linear dynamics with adaptive critical damping (exponential convergence related to the fastest response with no overshooting), being perturbed by the overestimation $\rho_z s$ caused by the compensation of the unknown term. The roots of the perturbed solution of (54) are given by

$$s_{1,2}^* = \frac{\alpha}{2} \pm j\sqrt{\rho_z(t)} \tag{57}$$

A condition of the following form can be used to avoid chattering (high frequency oscillations caused by a large imaginary value in the pole position as a result of overestimation) at the steady-state response.

$$|\kappa|z|^{\zeta+1}\,\text{sign}(s) + \dot{d}(t)| \le \kappa|z|^{\zeta+1} + \Delta < \rho_z^{max} \tag{58}$$

This provides an upper bound of the perturbation generated at the dynamics with respect to the solution with $\kappa = 0$ and $d(t) = 0$. In order to estimate the correlation between the sampling time $\tau$ and the natural frequency $\sqrt{\rho_z^{max}}$ (in $\frac{rad}{s}$), we must verify that the frequency given by the *Nyquist-Shannon* sampling theorem (the maximum operating frequency for a system with sampling time $\tau$) does not create a change in sign in $s(t)$ at the limit condition $|z(t)| = \mu$, that is

$$\sqrt{\rho_z^{max}} \le \frac{\pi}{\tau}\mu \tag{59}$$

Applying condition (58) an upper bound for $\kappa$ as a function of $\Delta$, $\tau$ and the absolute value of $z(t)$ is obtained as

$$\kappa \leq \frac{(\frac{\pi\mu}{\tau})^2 - \Delta}{|z|^{\zeta+1}} \qquad (60)$$

This constraint provides a limit on the application of the method that employs the bound $\Delta$ of the disturbance derivative, the sampling time $\tau$ and, taking into consideration relation (27), the required precision $\mu$.

Inside $\Omega_z$ we have that

$$\left| s(t) + \frac{\alpha}{2}\bar{s}(t) \right| < \mu$$

which geometrically entails:

$$|\bar{s}(t)| < \frac{\mu}{|\sin(\vartheta)|} \qquad (61)$$

$$|s(t)| < \frac{\mu}{|\cos(\vartheta)|} \qquad (62)$$

with $\vartheta$ defined in (32). Inside $\Omega_s$ we have that

$$|\dot{e}(t) + \lambda e(t)| < \frac{\mu}{|\cos(\vartheta)|}$$

Following the previous approach implies that:

$$|e(t)| < \frac{\mu}{|\cos(\vartheta)||\sin(\theta)|} \qquad (63)$$

$$|\dot{e}(t)| < \frac{\mu}{|\cos(\vartheta)||\cos(\theta)|} \qquad (64)$$

with $\theta$ defined in (31).

Applying Theorem 1 with

$$\phi_1(||e||) = k_1 e^2$$
$$\phi_2(||e||) = k_2 e^2$$
$$\phi_3(||e||) = k_3 |e|$$
$$W(e) = k_4 e^2$$
$$V(e) = \frac{1}{2} e^2$$
$$\zeta(e) = \frac{k_5}{k_4 |e|}$$

where $k_1 < 0.5$, $k_2 > 0.5$, $k_3 > 1.0$, $k_4 < \lambda$ and $k_5 > 1.0$, entails that cascade system given in (33) and (34) is GUAS with respect to the closed compact sets $\Omega_e$ and $\Omega_s$, respectively. □

## 4. Numerical Simulations

In this section we introduce numerical simulations of the course keeping problem with parameters given in Table 1 and being executed under the following assumption.

**Assumption 4.** *The numerical simulations are executed using the explicit Euler method with fixed sampling time $\tau = 0.1$ s.*

**Table 1.** Model parameters.

| Parameter | Value |
|:---:|:---:|
| $K$ | 0.21 |
| $T$ | 107.76 |
| $a_1$ | 13.17 |
| $a_2$ | 16,323.46 |

### 4.1. Constant Yaw Reference

This test is presented in [12] with a required a change in the yaw orientation angle from zero initial condition up to 50 degrees assuming that $d(t) = 0$. Table 2 show the parameters used in [12].

**Table 2.** Nonlinear concise backstepping controller parameters.

| Parameter | Value |
|:---:|:---:|
| $k_1$ | 0.0017 |
| $\omega$ | 0.6000 |

Based on this results the parameter $a_2$ of the synergetic controller presented in [10] is changed to achieve the same MIA at the end of the simulation. Table 3 show the parameters used with this algorithm.

**Table 3.** Synergetic controller parameters.

| Parameter | Value |
|:---:|:---:|
| $a_1$ | 0.090 |
| $a_2$ | 1.891 |
| $T_1$ | 28.000 |

The parameters of the AISM algorithm are obtained as follows

- Consider a settling time $t_s = 150s$, a maximum desired yaw rate $r_{max} = \frac{0.70\pi}{180}$ degrees per second and a required precision $\mu = 1.0 \times 10^{-6}$.
- The value of $\alpha(0)$ is obtained assuming an exponential convergence of the error from initial condition $e(0)$ to desired precision $\mu$ with a desired settling time $t_s$

$$\alpha(0) = -1.25 \frac{log\left(\frac{\mu}{|e(0)|}\right)}{t_s} = 0.0756 \tag{65}$$

The values of $\alpha_{min}$ and $\alpha_{max}$ are selected as

$$\alpha_{min} = \alpha(0) \tag{66}$$

$$\alpha_{max} = 5(\alpha_{min} + \Delta) \tag{67}$$

- The value of $\lambda_{min}$ is related to the initial conditions of the problem and the maximum desired yaw rate as

$$\lambda_{min} = \frac{r_{max}}{|e(0)|} = 0.014 \tag{68}$$

and $\lambda_{max}$ is calculated as

$$\lambda_{max} = 2.0\lambda_{min} = 0.028 \tag{69}$$

- The value of $\kappa$ must be higher than $\Delta$ in order to obtain a small value for $\mu$. Due ti the low-pass filtering properties of (54), the value of $\Delta$ can be further refined by estimating the cut-off frequency $\omega_c(t)$ of the second order system related to $s(t)$

$$\omega_c(t) = \frac{\alpha(t)}{2} \tag{70}$$

Therefore $\kappa$ is calculated as an adaptive gain that takes account of $\omega_c$ and the desired precision

$$\kappa(t) = \frac{\alpha(t)}{2\mu^{\frac{1}{\zeta+1}}} \tag{71}$$

- Simulations are used to set the values of $\zeta_{min}$ and $\zeta_{max}$ such that the value of the performance index MIA is equal to the value achieved with benchmark chosen controllers at the conclusion of the test period.

$$\zeta_{min} = 0.800$$
$$\zeta_{max} = 1.685$$

This condition generates an adequate adaption of the value of $\zeta$ that allows to obtain the desired power factor profile with respect to the absolute value of $e(t)$.

Figure 3 shows the states and control effort, and it can be seen that all of the solutions have a comparable setting time. The yaw error evolves similarly in all circumstances, however AISM can achieve the same high accuracy in steady state than Synergetic control with less control effort and a lower maximum yaw rate.

Figure 4 depicts the progression of performance indices over time, with a detailed view of the MIA performance index at the end of the test and final numerical values in Table 4. The evolution of the adaptive parameters employed in the proposed AISM method is detailed in Figure 5. In the case of $d(t) = 0$, the value of the performance parameter $\lambda$ is adjusted to match the intended low/high gain profile, while the value of gain $\alpha$ grows until it reaches the sliding condition, then falls to its lower bound.

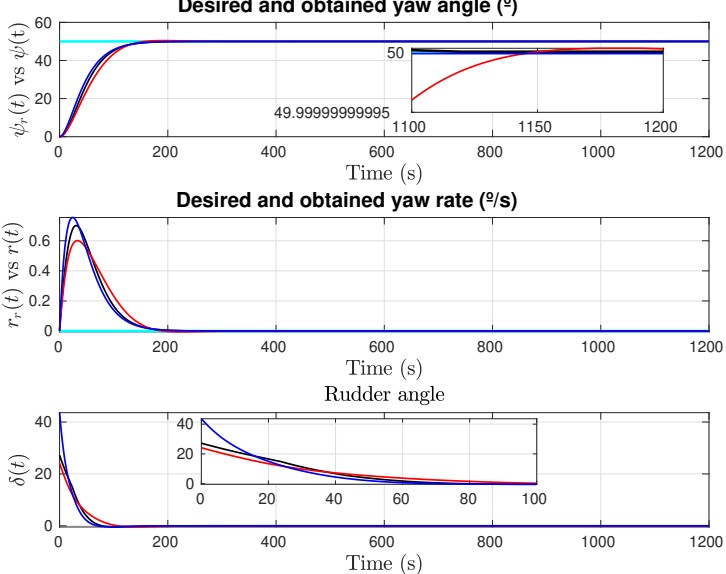

**Figure 3.** Constant yaw reference test with $d(t) = 0$. States and control. Cyan line: Reference; Red line: Concise backstepping (Zhang et al.); Blue line: Synergetic (Muhammad et al.); Black line: Adaptive sliding mode (González-Prieto et al.).

**Table 4.** Constant yaw reference test with $d(t) = 0$. Performance indices.

| Algorithm | MAE | MIA | MTV |
|-----------|-----|-----|-----|
| Concise Backstepping [12] | 0.042227 | 0.011348 | $3.6396 \times 10^{-5}$ |
| Synergetic [10] | 0.035641 | 0.011348 | $6.4436 \times 10^{-5}$ |
| AISM | 0.038468 | 0.011348 | $4.0722 \times 10^{-5}$ |

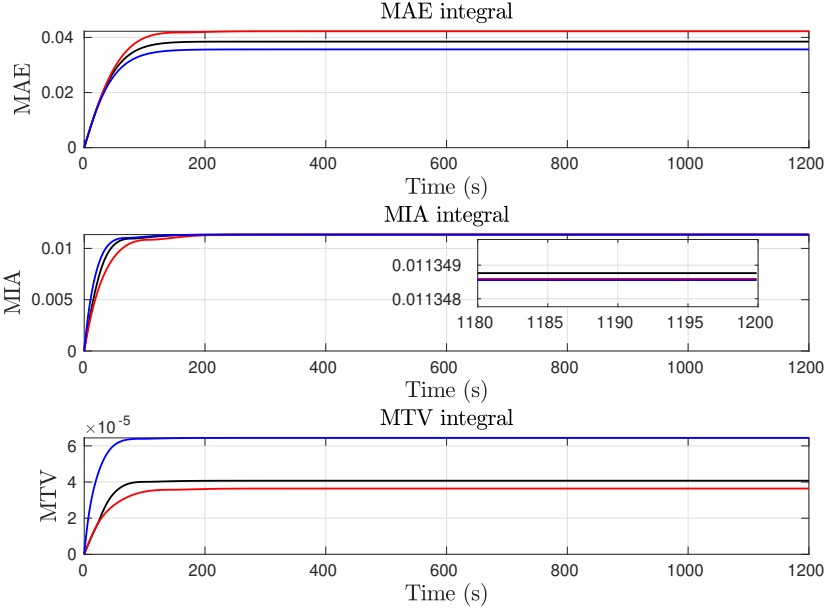

**Figure 4.** Constant yaw reference test with $d(t) = 0$. Performance indices evolution. Cyan line: Reference; Red line: Concise backstepping (Zhang et al.); Blue line: Synergetic (Muhammad et al.); Black line: Adaptive sliding mode (González-Prieto et al.).

Next, in order to test the robustness of the algorithms, the following disturbance is considered in (4)

$$d(t) = D[\cos(\omega_d t) + 0.83 \sin(3.29\omega_d t - 0.14)$$
$$+ 1.23 \cos(8.12\omega_d t + 0.26)$$
$$+ 0.65 \sin(1.37\omega_d t + 0.36)e^{\cos(2.21\omega_d t + 0.13)}] \tag{72}$$

with

$$D = 0.0025$$

$$\omega_d = 0.0703 \frac{rad}{s}$$

Figure 6 shows the states and control effort, and it is obvious that the suggested AISM negates the influence of the external disturbance, maintaining the intended performance at steady-state, and creating a rudder angle control that enables quick reaction attenuation without causing overshooting. The evolution of the sliding variable $s(t)$ and the external disturbance $d(t)$, introduced to evaluate the resilience qualities of the compared algorithms, is depicted in Figure 7.

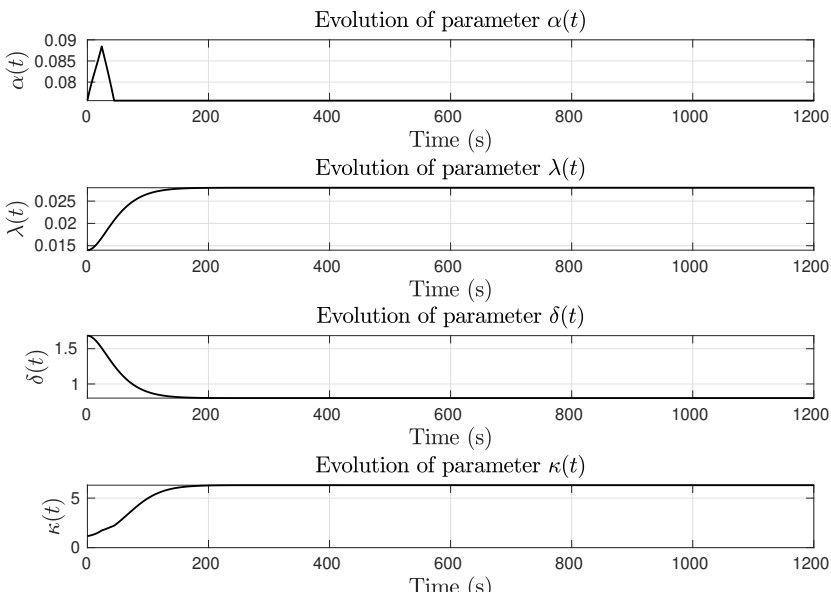

**Figure 5.** Constant yaw reference test with $d(t) = 0$. Adaptive parameters evolution.

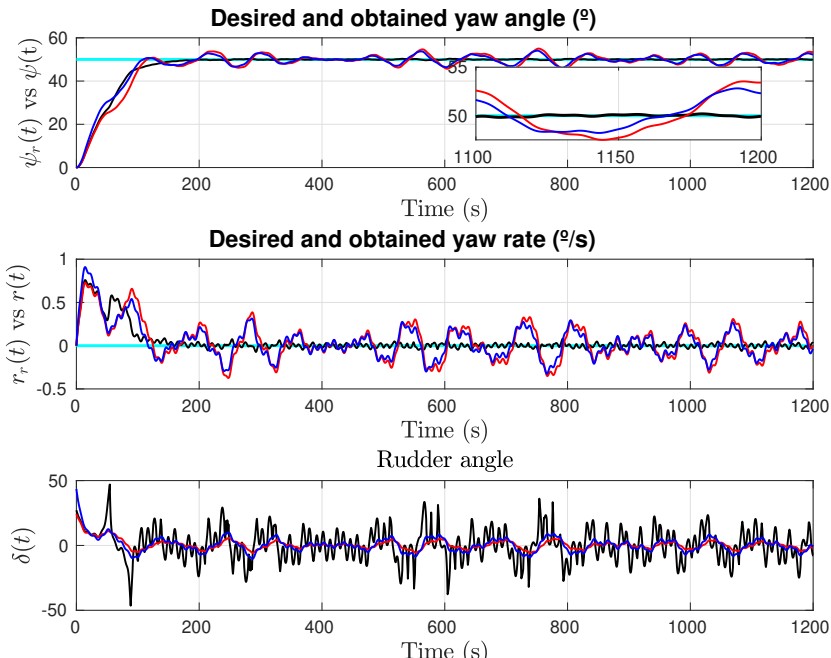

**Figure 6.** Constant yaw reference test with $d(t) \neq 0$. States and control. Cyan line: Reference; Red line: Concise backstepping (Zhang et al.); Blue line: Synergetic (Muhammad et al.); Black line: Adaptive sliding mode (González-Prieto et al.).

The evolution of the adaptive gains is seen in Figure 8. It is evident that the influence of the unknown disturbance is passed to $\alpha$ and $\kappa$ in order to maintain steady-state performance, but it has no effect on lambda or the adaptive power factor delta. As a result, the reaction achieves a quick response to external disturbances in order to maintain a constant target performance related to the sliding condition $s(t) \approx 0$.

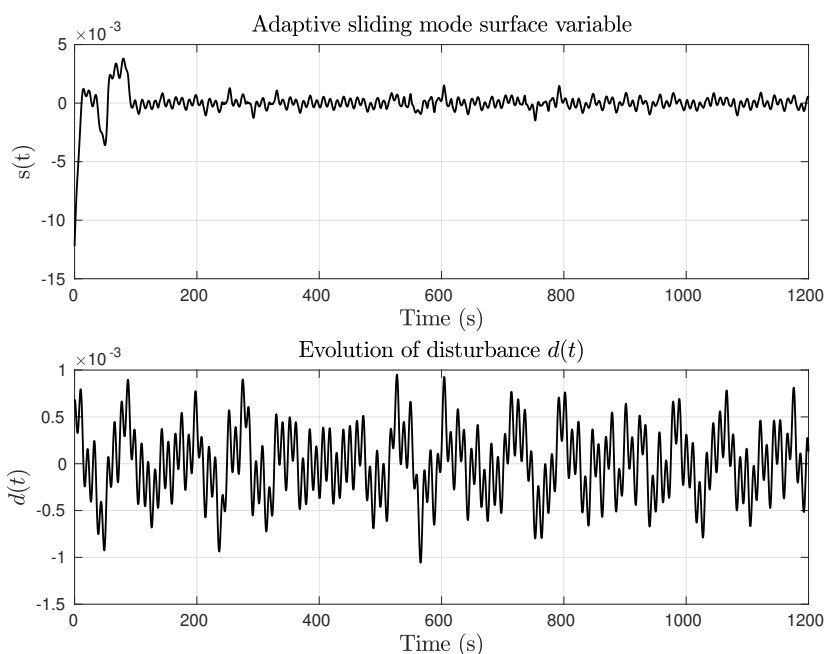

**Figure 7.** Constant yaw reference test with $d(t) \neq 0$. Sliding mode variable $s(t)$ and external disturbance $d(t)$.

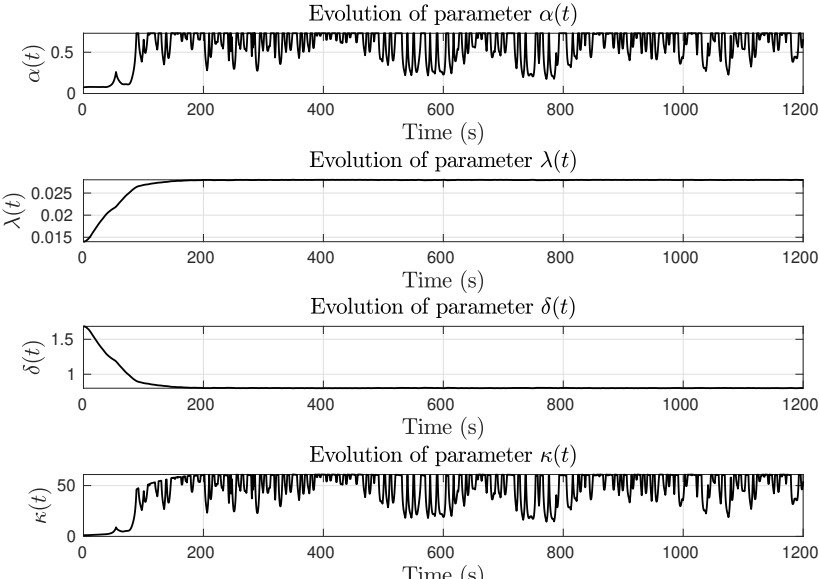

**Figure 8.** Constant yaw reference test with $d(t) \neq 0$. Adaptive parameters evolution.

*4.2. Sinusoidal Yaw Reference*

In this case, as in [12], the yaw reference to follow is a sinusoidal signal defined as

$$\omega_d = \frac{50\pi}{180.0} \sin(\frac{2\pi}{600.0}) \tag{73}$$

where the initial yaw angle is

$$\psi(0) = \frac{10\pi}{180.0} \tag{74}$$

States and control effort are provided in Figure 9 where it is clear that AISM is capable to follow the yaw reference with no appreciable delay keeping the desired settling time,

and Figure 10 depicts the adaptive parameter's change over time in the unperturbed case with sinusoidal reference.

As in the previous test, results with sinusoidal reference are tested introducing disturbance (72). Figure 11 shows the states and control effort obtained in this case, where, as in the constant reference test, the steady-state performance and the settling time obtained with AISM are preserved despite the presence of the external unknown disturbance.

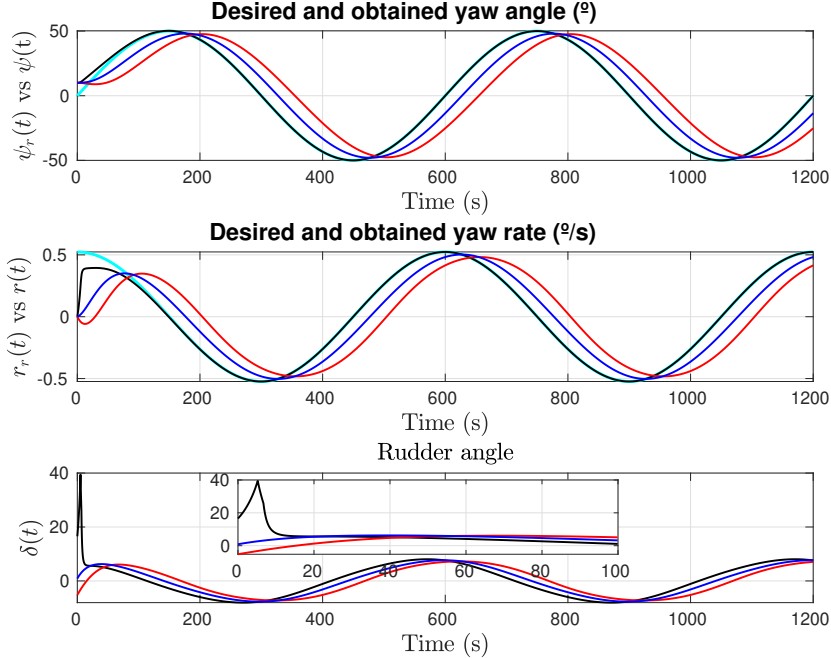

**Figure 9.** Sinusoidal yaw reference test with $d(t) = 0$. States and control. Cyan line: Reference; Red line: Concise backstepping (Zhang et al.); Blue line: Synergetic (Muhammad et al.); Black line: Adaptive sliding mode (González-Prieto et al.).

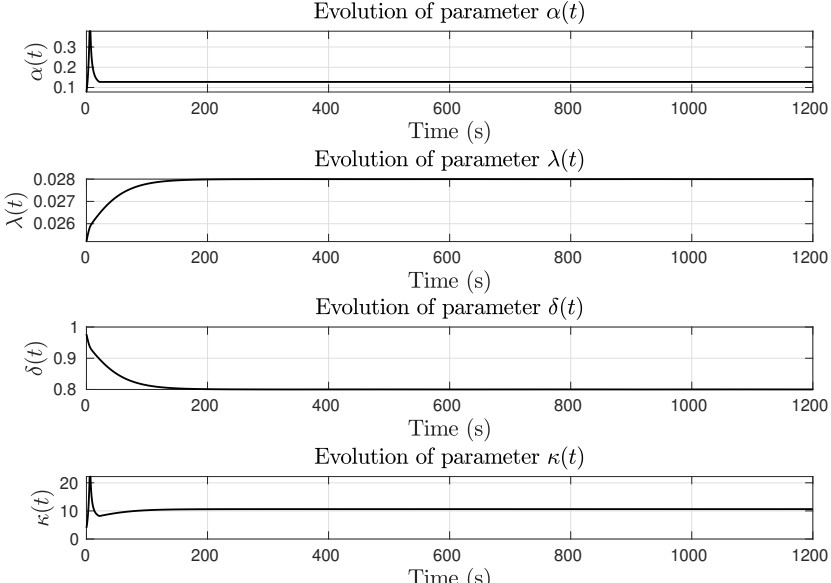

**Figure 10.** Sinusoidal yaw reference test with $d(t) = 0$. Adaptive parameters evolution.

Figure 12 shows the evolution of the sliding variable and the external disturbance input $d(t)$. The time evolution of adaptive parameters in Figure 13 mirrors the behavior in the case of constant yaw reference in case of $d(t) = 0$. This characteristic, like in constant yaw reference scenario, generates quick reactions to external disturbances, maintaining a desired fixed performance and avoiding a delayed response.

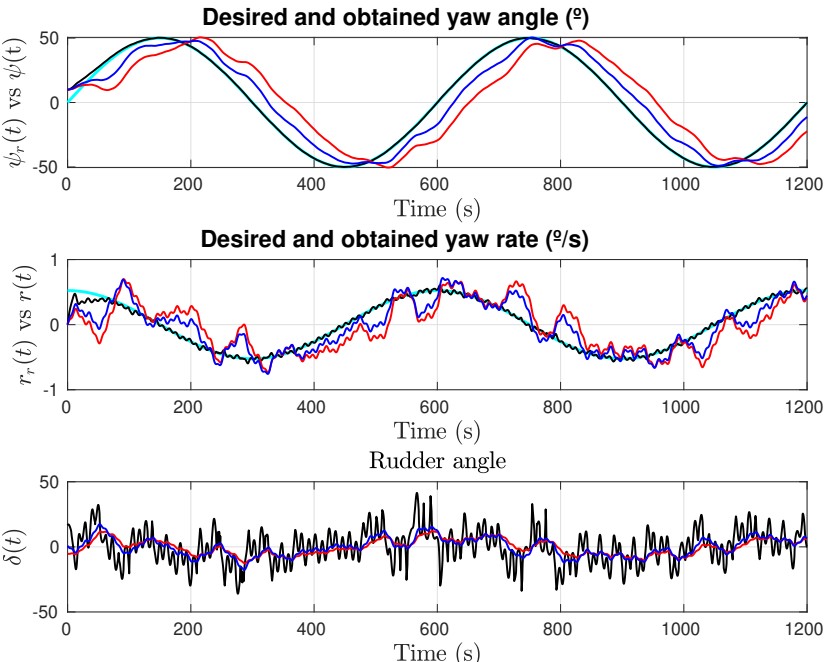

**Figure 11.** Sinusoidal yaw reference test with $d(t) \neq 0$. States and control. Cyan line: Reference; Red line: Concise backstepping (Zhang et al.); Blue line: Synergetic (Muhammad et al.); Black line: Adaptive sliding mode (González-Prieto et al.).

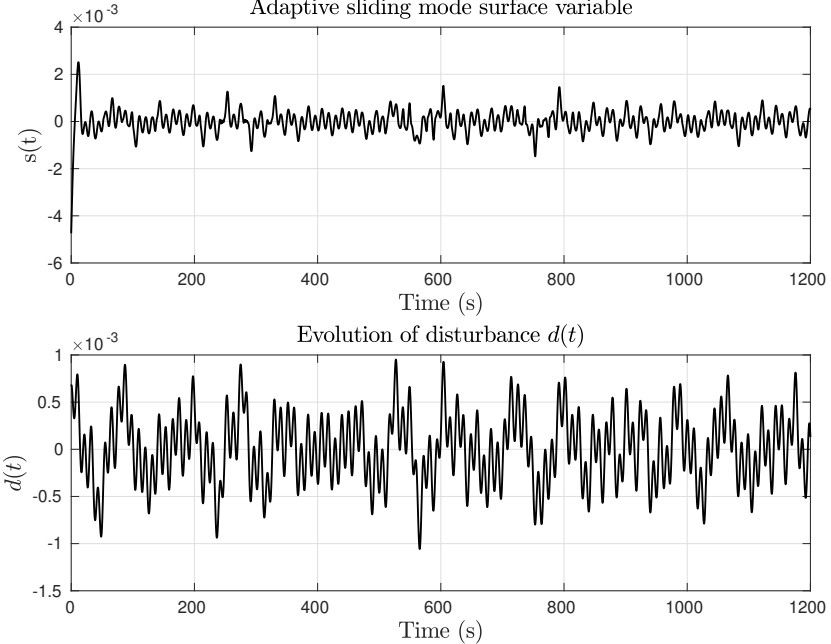

**Figure 12.** Sinusoidal yaw reference test with $d(t) \neq 0$. Sliding mode variable $s(t)$ and external disturbance $d(t)$.

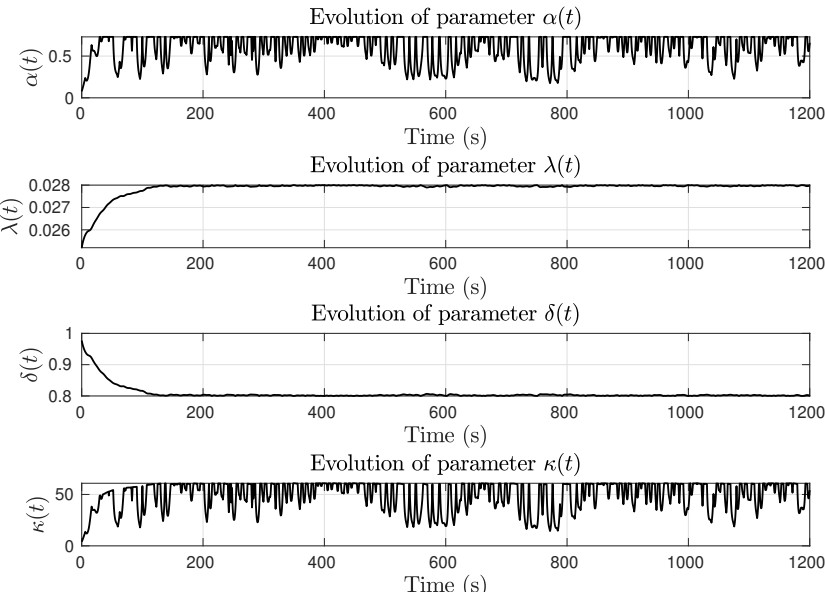

**Figure 13.** Sinusoidal yaw reference test with $d(t) \neq 0$. Adaptive parameters evolution.

## 5. Conclusions

We present an approach to developing an adaptive integral sliding mode procedure to design a nonlinear controller for the course keeping of surface vehicles. A solution has been proposed that is based on the application of adaptive gains that change the sliding surface's dumping properties, resulting in a low/high gain profile so that it can overcome the need for large control inputs at initial conditions while achieving a higher gain at steady state.

Results obtained in numerical simulations demonstrate that the proposed AISM algorithm achieves the desired performance with fixed and time-varying references that cancel out external disturbances. A fixed parameter configuration is used to evaluate the performance based on the settling time, maximum allowable yaw rate, and steady state precision. Due to its robustness, the algorithm achieves the desired response without requiring the development of a new parameter configuration for each type of test.

An advantage of the method is its robustness with respect to an overestimation of $\Delta$: the performance is not highly degraded if this bound is not accurately known. However, choosing an overly large value might cause oscillations in the response of the estimation error.

In order to integrate an optimal point of view in the design of the adaptive parameters, a deepest study of the function that can determine the adaptive values of $\lambda(e)$ and $\zeta(e)$ is an interesting open problem that can be discussed from the standpoint of model predictive control.

The extension of this procedure with the assumption of partial state feedback will be addressed in future researches by means of the application of an adaptive integral sliding mode observers.

**Author Contributions:** J.A.G.-P.: Conceptualization, Methodology, Software, Formal analysis, Writing—original draft, C.P.-C.: Formal analysis, Writing—original draft, Y.S.: Formal analysis, Writing—original draft. All authors have read and agreed to the published version of the manuscript.

**Funding:** J. A. González-Prieto and C. Pérez-Collazo acknowledges funding from the Defense University Center at the Spanish Naval Academy, Spanish Ministry of Defense, TRABODIT project (PICUD-2021-01).

**Institutional Review Board Statement:** Not applicable.

**Informed Consent Statement:** Not applicable.

**Data Availability Statement:** Not applicable.

**Conflicts of Interest:** The authors declare that they have no known competing financial interests or personal relationships that could have appeared to influence the work reported in this paper.

**Abbreviations**

The following abbreviations are used in this manuscript:

| | |
|---|---|
| USV | Unmanned Surface Vehicle |
| IMO | International Maritime Organisation |
| LOS | Line of Sight |
| SMC | Sliding Mode Control |
| AISM | Adaptive Integral Sliding Mode |
| GUAS | globally uniformly asymptotically stable |
| MAE | Mean Absolute Error |
| MIA | Mean Integral Absolute |
| MTV | Mean Total Variation |

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
