# Peer review of "Adaptive Integral Sliding Mode Based Course Keeping Control of Unmanned Surface Vehicle"

_jmse, doi:10.3390/jmse10010068_

Round 1

Reviewer 1 Report

In this paper, the path following control of the USV was investigated under the unknown disturbances and system uncertainties. Tough many advanced algorithms have been presented in the existing literatures, there still exists some challenges, which should be further addressed. Therefore, the proposed adaptive slide mode control algorithm is effective to keep the yaw error within the desired boundaries. In addition, in my opinion, the topic in the manuscript is valuable and original that can give significant value to the audience, especially for the path following control. After reviewing the manuscript, the following points are my comments, and I hope that can be tackled by the authors to improve the quality of their manuscript.

1) In the manuscript, we can find that the proposed control law (Eq. (21)), the system uncertainties were included. If the authors assumed that system uncertainties were known? Please add the related expansions.

2) Some grammar mistakes should be checked and revised. Such as, in page 3, “v the sway velocity, r the yaw rate, Iz the rotational inertia …”

3) For the path following of the USV, the detailed literature reviews have been introduced in this manuscript. However, as fa as I known, the event-triggered control has been introduced to the unmanned surface vessels, and it is with obvious advantages to reduce the transmission burden and it also can avoid the effect of the low frequency disturbance. For example, Robust Adaptive Neural Control for Wing-Sail-Assisted Vehicle via the Multiport Event-Triggered Approach. I suggest the authors add the corresponding description to perfect the section 1.

4) To facilitate the understanding of the readers, the main contributions and the conclusion should be reworded. And some remarks should be added illustrated the effectiveness and the robustness of the proposed adaptive slide mode technique. 

5) English needs to be polished. Besides, the manuscript should be formatted better, especially for the format of the references.

In brief, this paper is satisfied to be published in the JMSE. Thus, I suggest this paper accept after minor revisions.

Reviewer 2 Report

Overall, the manuscript is well written and does fit the scope of JMSE. In my opinion, it is a study to deal with a necessary problem, however, there are some things that need to be addressed to meet the quality publication. Some of the concerns are as follows:

1/ Despite the motivating topic, the theoretical contribution of the manuscript does not seem significant and the experimental contribution does not consider. Thus, I am concerned about the application and practicability of this theory for the unmanned surface vehicle. This paper should have more theoretical contributions. The novelty of the approach is rather low since it seems all the methods are existing.

2/ The title of the paper should be changed. The contribution of the paper is the heading control or course keeping of USV.  The paper is simply a simulated yaw model in a second-order form that is not related to the 3DOF of the USV system. The path following control in this paper is very simple without considering the path planning algorithm and surge velocity control.

3/ The motivation and background of wide practical use of the theoretic results presented should be clearly emphasized to facilitate the readers.

4/ Besides, some new relative references have to be cited in relation to the current work on DP control and sliding mode control of marine systems: Station-Keeping Control of a Hovering Over-Actuated Autonomous Underwater Vehicle Under Ocean Current Effects and Model Uncertainties in Horizontal Plane (IEEE Access, 2021); Design of a Non-Singular Adaptive Integral-Type Finite Time Tracking Control for Nonlinear Systems With External Disturbances (IEEE Access, 2021); Perturbation Observer-Based Robust Control Using a Multiple Sliding Surfaces for Nonlinear Systems with Influences of Matched and Unmatched Uncertainties (Mathematics, 2020), Fast Terminal Sliding Control of Underactuated Robotic Systems Based on Disturbance Observer with Experimental Validation (Mathematics, 2021).

5/ A control system block diagram should be given for the proposed control scheme.

6/ If the Lyapunov functions are chosen via the viewpoint of practical application, the authors should give some effective suggestions. More discussions should be given to clearly demonstrate the limitations/validity of the obtained results.

7/ In section 4.2 of simulation results, more simulation results need to be added in the paper, such as the performance indices evolution, and adaptive parameters evolution (like Figures 3 and 4 in section 4.1). Moreover, more discussions should be given to clearly demonstrate the effectiveness of the obtained results.

8/ The analysis in this paper should be supported by experimental results. The authors should use practical systems to validate the proposed methods with experiment results. The validity of these relevant to applications is impossible to judge without experimental testing. 

9/ The English writing of this paper should be thoroughly polished, especially some grammatical errors and formula format errors are required to be revised carefully.

Round 2

Reviewer 2 Report

Thank you for the revised manuscript. The new version is good now. The quality of the paper has been improved by properly addressed my previous comments. For this, the paper is much better structured and easy to understand.

Other minor comments:

In ref [50] and [53], the author´s name is wrong. Please re-arrange as below:

Alattas, K.A., Mobayen, A., Din, S.U., Asad, J.H., Fekih, A., Assawinchaichote, W., Vu, M.T., 2020. Design of a Non-Singular Adaptive Integral-Type Finite Time Tracking Control for Nonlinear Systems With External Disturbances. IEEE Access, Volume 9, pp. 102091-102103. DOI: 10.1109/ACCESS.2021.3098327

Thanh, H.L.N.N.; Vu, M.T.; Mung, N.X.; Nguyen, N.P.; Phuong, N.T. Perturbation Observer-Based Robust Control Using a Multiple Sliding Surfaces for Nonlinear Systems with Influences of Matched and Unmatched Uncertainties. Mathematics 2020, 8, 1371. https://doi.org/10.3390/math8081371

In addition, the format of references needs to be unified with the aim to satisfy the requirement of the journal, and the DOI number of some new references needs to be added in this paper. Please check carefully.